# Preliminary Biological Activity Screening of *Plectranthus* spp. Extracts for the Search of Anticancer Lead Molecules

**DOI:** 10.3390/ph14050402

**Published:** 2021-04-23

**Authors:** Epole Ntungwe, Eva María Domínguez-Martín, Catarina Teodósio, Silvia Teixidó-Trujillo, Natalia Armas Capote, Lucilia Saraiva, Ana María Díaz-Lanza, Noélia Duarte, Patrícia Rijo

**Affiliations:** 1CBIOS—Universidade Lusófona Research Center for Biosciences & Health Technologies, Universidade Lusófona de Humanidades e Tecnologias, Campo Grande 376, 1749-024 Lisboa, Portugal; ntungweepolengolle@yahoo.com (E.N.); evam.dominguez@uah.es (E.M.D.-M.); catarina.teodosio@gmail.com (C.T.); 2Department of Biomedical Sciences, Faculty of Pharmacy, University of Alcalá de Henares, Ctra. A2, Km 33.100—Campus Universitario, 28805 Alcalá de Henares, Spain; ana.diaz@uah.es; 3Centro Atlántico del Medicamento S.A., Avenida Trinidad 61, 7ª Planta, Torre Agustín Arévalo, 38204 La Laguna, Tenerife, Spain; teixido.silvia@gmail.com (S.T.-T.); nataliaarmas@ceamedsa.com (N.A.C.); 4LAQV/REQUIMTE, Laboratório de Microbiologia, Departamento de Ciências Biológicas, Faculdade de Farmácia, Universidade do Porto, Rua de Jorge Viterbo Ferreira n.º 228, 4050-313 Porto, Portugal; lucilia.saraiva@ff.up.pt; 5Instituto de Investigação do Medicamento (iMed.ULisboa), Faculdade de Farmácia, Universidade de Lisboa, 1649-003 Lisboa, Portugal; mduarte@ff.ulisboa.pt

**Keywords:** *Plectranthus*, royleanone, 7α-acetoxy-6β-hydroxyroyleanone, *ent*-abietane, antitumoral activity, antimicrobial activity, antioxidant

## Abstract

*Plectranthus* species (Lamiaceae) have been employed in traditional medicine and this is now validated by the presence of bioactive abietane-type diterpenoids. Herein, sixteen *Plectranthus* acetonic extracts were prepared by ultrasound-assisted extraction and their biological activity was screened. The antimicrobial activity of each extract was screened against yeasts, and Gram-positive and Gram-negative bacteria. The *P. hadiensis* and *P. mutabilis* extracts possessed significant activity against *Staphylococcus aureus* and *Candida albicans* (microdilution method). Moreover, all extracts showed antioxidant activity using the DPPH method, with *P. hadiensis* and *P. mutabilis* extracts having the highest scavenging activities. Selected by the *Artemia salina* model, *P. hadiensis* and *P.ciliatus* possessed low micromolar anti-proliferative activities in human colon, breast, and lung cancer cell lines. Furthermore, the most bioactive extract of *P. hadiensis* leaves and the known abietane diterpene, 7α-acetoxy-6β-hydroxyroyleanone isolated from this plant, were tested against the aggressive type triple negative breast cancer (MDA-MB-231S). *P. hadiensis* extract reduced the viability of MDA-MB-231S cancer cell line cells, showing an IC_50_ value of 25.6 µg/mL. The IC_50_ value of 7α-acetoxy-6β-hydroxyroyleanone was 5.5 µM (2.15 µg/mL), suggesting that this lead molecule is a potential starting tool for the development of anti-cancer drugs.

## 1. Introduction

Over the centuries, plants and natural products derived from plants have been the basis of traditional medicine. Nowadays, plant-based medicines have been playing an important role in drug discovery and development. They are also widely employed in various public health practices as they are safe, cost-effective, and possess unique chemical diversity [1,2]. Cancer remains one of the leading causes of death globally, with approximately 18.1 million new cases and 9.6 million cancer-related deaths in 2018, according to the World Health Organization [3]. Furthermore, several types of chemotherapies used are ineffective and generate unwanted adverse side effects [4,5]. Similarly, there is an increasing number of cases of bacterial infections that are resistant to current antibiotics and are difficult or impossible to treat [6]. Currently, there is an emerging interest in developing drugs that overcome the problems stated above by using natural compounds.

*Plectranthus* L’ Herit. is a major genus of the Lamiaceae family comprising about 300 species mainly distributed in the summer-rainfall savannahs and forested regions of tropical Africa, Asia, and Australia [7,8]. Most of the *Plectranthus* species are soft trailing semi-succulent to succulent herbs or shrubs and their stems, leaves, roots, and tubers are frequently used as traditional medicines for the treatment of various illnesses, including respiratory, digestive, and liver ailments [9]. Phytochemical studies on some species of *Plectranthus* revealed the presence of a large number of diterpenes and triterpenes [10,11]. Isolated terpenes from *Plectranthus* species are reported to possess antibacterial [12,13,14], antitumoral [15,16,17,18], antifungal [12,19], insecticidal [20], and antiplasmodial [20] activities. Moreover, our group has been focused on the phytochemical study of *Plectranthus* species and we have reported abietane diterpenoids with diverse bioactivity [16,17,21,22,23,24,25]. Therefore, the screening of other *Plectranthus* spp. extracts aiming at finding new sources of biologically active natural products is warranted.

Herein, sixteen *Plectranthus* species were screened for their bioactivity (antioxidant, antimicrobial activities, and general toxicity) and the main compound of the most bioactive extract was identified. To the best of our knowledge, the scientific literature concerning these species is scarce or even non-existent. *P. hadiensis* was earlier reported to have ethnopharmacological activity and to be a rich source of ent-abietane diterpenes [11]. This work aims to screen the antimicrobial, antioxidant, and cytotoxic properties of several *Plectranthus* spp. extracts and identifying the component in the most bioactive extract that may be responsible for its bioactivity.

## 2. Results and Discussion

All sixteen *Plectranthus* spp. extracts were prepared using ultrasound-assisted extraction in acetone. Previous studies of *Plectranthus* species showed that acetonic extracts are rich in diterpenoids, and high extraction yields are obtained when the ultrasound extraction method is carried out [21]. Therefore, using this extraction method, sixteen acetonic extracts of *Plectranthus* spp. were prepared. The extraction of all extracts was done in triplicate under the same conditions. All extracts were solubilized in DMSO (10 mg/mL) and stored at −20 °C until further analysis for their biological assays. *P. mutabilis* had the highest extraction yield (30.00% *w*/*w*) (Table 1).

The antioxidant activity of the extracts was quantitatively determined using the DPPH (2,2-diphenyl-1-picrylhydrazyl) scavenging radical assay. The antioxidant activity of the extracts was compared with quercetin, a well-known pure compound used as a positive control to understand the potential antioxidant activity of the extracts screen. The antioxidant activity indicates the capacity to quench reactive oxygen species (ROS), leading to decreased oxidative stress [26]. The results of the free radical scavenging capacity of extracts at a concentration of 10 µg/mL are shown in Table 1. *P. mutabilis* and *P. hadiensis* had the highest scavenging activity of 46.14% and 36.24%, respectively. These results are also in agreement with other studies with *Plectranthus* extracts, in which *P. madagascarensis P. neochilus*, *P. barbatus* and *P. verticillatus* extracts showed free radical scavenging abilities [27]. This could be due to the presence of abietane diterpenes known for their antioxidant activity. Recently, findings concerning the antioxidant activity of abietane diterpenes isolated from *Plectranthus* spp. suggest that the quinone moiety present in these compounds is probably responsible for their biological activity [28]. The presence of the 12-OH group and the carbonyl group at position C-7 (*p*-position) could serve as hydrogen- and/or electron-donating moieties, resulting in the formation of stable quinone derivatives [29].

To evaluate the antimicrobial activity, the extracts were screened against Gram-positive (*Staphylococcus aureus*, *Enterococcus faecalis*) and Gram-negative (*Pseudomonas aeruginosa* and *Escherichia coli*) bacteria and yeasts (*Candida albicans* and *Saccharomyces cerevisiae*) using the well diffusion assay. The antimicrobial activity of each acetonic extract was first screened by the well diffusion method at a concentration of 1 mg/mL. Only *P. hadiensis* and *P*. *mutabilis* extracts showed antimicrobial activity against *S. aureus* with inhibition zones of 16 mm and 15 mm, respectively. None of the extracts showed significant antibacterial activity against *E. faecalis* and Gram-negative bacteria. These results are in agreement with previous works on *Plectranthus* spp. [17,30] showing that only Gram-positive bacteria are sensitive to *Plectranthus* acetonic extracts [27,31]. When the zone of inhibition of the extracts was compared with the negative control, it was found that nine extracts (*P. ciliatus*, *P. welshii, P. mzumbulensis*, *P. inflexus*, *P. lucidus*, *P. xerophylus*, *P. lippio*, *P. hadiensis*, and *P. mutabilis*.) exhibited antimicrobial activity against *C. albicans (10–11 mm)*. The remaining did not display any antimicrobial effects, showing inhibition zones similar to the negative control (data not shown).

As the initial screening using the well diffusion assay identified *P. hadiensis* and *P. mutabilis* extracts as possessing the most promising antimicrobial activities, further studies were carried out to evaluate the minimum inhibitory concentration (MIC) and minimum bactericidal concentration (MBC) or minimum fungicidal concentration (MFC) against the susceptible strains (Table 2). The MIC values ranged from 3.91 µg/mL to 125 µg/mL. *P. hadiensis* was the most active extract, exhibiting a MIC value of 3.91 µg/mL against the methicillin-resistant *S. aureus* strain (MRSA), similar to that observed for the positive control (1.95 µg/mL). The MBC/MFC values ranged from 62.5 to 250 µg/mL. It is possible to conclude that the extracts are mainly bacteriostatic rather than bactericidal [32].

To identify the most promising extracts with cytotoxic activity, the screening of general toxicity using the *Artemia salina* model was carried out. This assay was employed to screen the sixteen extracts because it is low-cost, rapid, convenient, and requires only a relatively small amount of sample [31,33]. All of the acetonic extracts were tested at 10 µg/mL with values ranging from 23.42 to 65.88% mortality (see Table 1). The most active extracts were further studied to obtain the LC_50_ values at concentrations of 0.1, 0.5, and 1 mg/mL, after 24 h of exposure. The most toxic extracts (*P. mutabilis*, *P. swynnertonii*, *P. hadiensis*, *P. ciliatus*, and *P. cylindraceus*) with LC_50_ ≤ 1 µg/mL (see Table 1) were then evaluated on human-derived cancer cell lines.

The cytotoxic activity was determined by the sulforhodamine B (SRB) assay in three different cancer cell lines: colon colorectal carcinoma (HCT116), human breast adenocarcinoma (MCF-7), and lung cancer carcinoma (NCI-H460). The IC_50_ values in all the cell lines tested ranged from 2.25 µg/mL to 36 µg/mL (Table 3). According to the National Cancer Institute (NCI), crude extracts that possess IC_50_ ≤ 500 µg/mL are of potential interest for further studies [34] and a possible candidate for further development of cancer therapeutic agents. Thus, most of the selected extracts showed potential values, and *P. hadiensis* and *P. ciliatus* seem to be potential sources of lead anticancer molecules. *P. ciliatus* extract was most active against the colon cell line (HCT116), whereas the *P. hadiensis* extract was the most active against the breast (MCF-7) and lung (NCI-H460) cell lines with the lowest IC_50_ value in the MCF-7 cell lines. For this reason, this extract was further tested against the aggressive type triple-negative breast cancer (MDA-MB-231S). MCF-7 hormone receptors expressing breast cancers have a more favorable prognosis as opposed to triple-negative breast cancer (TNBC), which is characterized by a poor treatment outcome [35]. This cell line is a highly metastatic triple-negative breast cancer cell line that does not display estrogenic receptors (ER), progesterone receptors (PR), or human epidermal growth factor receptor 2 (HER2), and is thus difficult to treat [21]. *P. hadiensis* acetonic extract had a growth inhibition effect on MDA-MB-231 cancer cells (IC_50_ value of 25.6 µg/mL, Table 3). Many studies have attributed the cytotoxicity of *Plectranthus* extracts to the presence of royleanone-type abietane diterpenoids with known anticancer activities [18,36]. The abietane-type diterpenoid royleanones are a highly bioactive group of lead molecules, important for the development of new anticancer drugs [22] Given the good levels of bioactivity of *P. hadiensis* extract in all the cell lines tested, it was selected to identify its main bioactive component.

To unveil the chemical profile of the most bioactive *P. hadiensis* acetonic extract, and to identify the main compound responsible for the tested bioactivity, an HPLC–DAD study was carried out. The chromatogram revealed that the known *ent*-abietane diterpene, 7α-acetoxy-6β-hydroxyroyleanone, Roy (Figure 1) was the major compound in the extract (Appendix A). To isolate this diterpene, a bio-guided column and the preparative chromatographic procedure were carried out. Its structure was confirmed through comparison of its spectroscopic data (Appendix A) to those described in the literature [12,29,37].

In previous studies, 7α-acetoxy-6β-hydroxyroyleanone showed cytotoxic activity against three human cell lines, namely, sensitive non-small cell lung carcinoma, NCI-H460 cell line (IC_50_ 2.7 ± 0.4), multidrug-resistant non-small cell lung carcinoma cell line with P-glycoprotein overexpression. NCI-H460/R (IC_50_ 3.1 ± 0.4), and human embryonal bronchial epithelial (MCR-5) cells (IC_50_ 8.6 ± 0.4) [21,38]. Given its cytotoxic properties, the presence of this compound in *P. hadiensis* acetonic extract may partially explain, the foreseen properties of the extract. To further explore its cytotoxicity, additional studies were performed.

The preliminary toxicity of the isolated 7α-acetoxy-6β-hydroxyroyleanone (Roy) was evaluated using the Brine shrimp lethality bioassay, which gave a percentage mortality of 30.95%. The cytotoxicity of 7α-acetoxy-6β-hydroxyroyleanone was further tested in the TNBC, MDA-MB231S cell line using the MTT assay. It had an IC_50_ value of 5.5 µM = 2.15 µg/mL (Appendix A), being approximately 12-fold more active than the corresponding extract (MDA-MB231S IC_50_ value = 25.6 µg/mL). This diterpene has also exhibited cytotoxic activity against breast, renal, melanoma, and central nervous system cancer cell lines [24,29,30]. It was also found to induce apoptosis in the H7PX glioma cell line, through the G2/M cell cycle arrest and DSBs (double-strand breaks) [39]. The cytotoxicity of 7α-acetoxy-6β-hydroxyroyleanone could be due to its royleanone-type scaffold and by its high lipophilicity, which facilitates penetration into the interior of the cell membrane [21,22,29,39]. Moreover, 7α-acetoxy-6β-hydroxyroyleanone was found to exhibit better activity against Gram-positive bacteria, and more importantly, against MRSA strains than some of the existing antibiotics [37]. Roy is found in many other *Plectranthus* species like *P. madagascariensis*, *P. grandidentatus*, *P. actites*, *P. amboinicus*, *P. sanguineus*, *P. argentatus*, and thus can be considered as a chemomarker of the *Plectranthus* genus [21,22].

## 3. Materials and Methods

### 3.1. Plant Material

All *Plectranthus* spp. studied in this work (Table 1) were grown in the “Parque Botânico da Tapada da Ajuda”, Lisbon-Portugal from cuttings provided by the Kirstenbosch National Botanical Gardens, South Africa. Plants were collected between 2007 and 2008, always in June and September. Voucher specimens were deposited in the Herbarium “João de Carvalho e Vasconcellos” of the Instituto Superior de Agronomia, Lisboa (LISI), Portugal. The plant names were verified with the Plant List [40].

#### Extraction Procedure

Plant extracts were obtained by the ultra-sonication method, adding 30 mL of acetone to 3 g of ground dry plants (*P. hadiensis* leaves and the whole plant for the remaining *Plectranthus* spp.), sonicated for 1 h, and filtered (Whatman No 5 paper, Inc., Clifton, NJ, USA). The extraction procedure was repeated three times until complete extraction [41]. The liquid samples were evaporated at 40–50 °C using a rotary evaporator (Sigma-Aldrich, IKA HBR 4 basic heating bath, Essen, Germany). All extracts were solubilized in DMSO (10 mg/mL, except for the exceptions that are mentioned) and stored at −20 °C until further analysis.

### 3.2. Phytochemical Study of P. Hadiensis

#### 3.2.1. HPLC-DAD Fingerprint Analysis

Extract profiling was performed with an Agilent Technologies 1260 Infinity II Series system with diode array detector (DAD; Agilent, Santa Clara, CA, USA), equipped with an Eclipse XDB-C18, (250 × 4.0 mm i.d., 5 µm) column, from Merck and ChemStation Software (Hewlett-Packard, Alto Palo, CA, USA). Four detection wavelengths were selected: 254, 270, 280, and 360 nm. The mobile phase consisted of a mixture of methanol (A), acetonitrile (B), and 0.3% (*w*/*v*) trifluoroacetic acid in ultrapure water (C). The employed method was modified from the one previously published Matías et al. [21] as follows: 0 min, 15% A, 5% B, and 80% C; 10 min, 70% A, 30% B, and 0% C; 25 min, 70% A, 30% B, and 0% C; and 28 min, 15% A, 5% B, and 80% C. The flow rate was set at 1 mL/min at room temperature and the injection volume was 20 µL. Solvents were previously filtered and degassed through a 0.22 μm membrane filter. The major peak from the *P. hadiensis* leaves extract was identified by co-elution, comparing the retention time and UV-vis spectrum overlayed with an authentic standard (Appendix A).

#### 3.2.2. Isolation and Structural Characterization of 7α-Acetoxy-6β-Hydroxyroyleanone

The *P. hadiensis* extract was fractionated by normal phase column chromatography over silica gel using mixtures of n-hexane: EtOAc (8: 2) as eluents to give 3 fractions (A, B, and C) in order of increasing polarity. Fraction B was further studied using preparative thin-layer chromatography (n-hexane/AcOEt 7:3) on pre-coated TLC sheets (Merck 7747, Darmstadt, Germany) giving 4 fractions B1 to B4. Visualization of spots was performed under visible light and UV light (λ 254 and 366 nm) followed by spraying with a mixture of H_2_SO_4_: AcOH: H_2_O (4:80:16) and heating. Fraction B2 was further purified by preparative chromatography affording its major compound, 7α-acetoxy-6β-hydroxyroyleanone. The NMR spectra were collected on a Bruker Fourier 300 spectrometer (^1^H 300 MHz, ^13^C 75 MHz) using CDCl_3_ as the solvent. ^1^H and ^13^C chemical shifts are expressed in δ (ppm) and the proton coupling constants (J) in hertz (Hz) Appendix A.

### 3.3. DPPH Radical Scavenging Assay

The free radical scavenging activity was measured by the DPPH method, as described by Rijo et al. [26]. Briefly, 10 μL of each extracted sample (1 mg of dry plant extract/mL) were added to a 990 μL solution of DPPH (0.002% in methanol). The mixture was incubated for 30 minutes in the dark, at room temperature. The absorbance (Abs) was measured at 517 nm (U-1500 Hitachi Instruments, Inc; USA). The positive control used was quercetin (10 mg/mL in methanol). An absorbance control (Abs_control_) containing 10 µL of methanol and 990 µL of DPPH was also prepared. Assays were carried out in triplicate and the free radical scavenging activity was calculated using Equation (1):(1)$$\mathrm{Scavenging}\;\mathrm{activity}\;\left(\%\right)=\frac{{\mathrm{Abs}}_{\mathrm{control}}-{\mathrm{Abs}}_{\mathrm{sample}}}{{\mathrm{Abs}}_{\mathrm{control}}}\times 100$$

### 3.4. Antimicrobial Screening Assays

#### 3.4.1. Microorganism Used

The microorganisms used in this study were obtained from the American Type Culture Collection (ATCC). They included five strains *Enterococcus faecalis* ATCC 29212, *Escherichia coli* ATCC 25922, *Pseudomonas aeruginosa* ATCC 27853, *Staphylococcus aureus* ATCC 25923, *Saccharomyces cerevisiae* ATCC 2601, *Staphylococcus aureus*. CIP were obtained from the CIP 106760, and the yeast strain *Candida albicans* ATCC 10231.

#### 3.4.2. Well Diffusion Method

The antimicrobial activity of each obtained extract was evaluated against two Gram-positive bacteria (*E. faecalis* and *S. aureus*), two Gram-negative bacteria (*E. coli* and *P. aeruginosa*), and two yeasts (*S. cerevisiae* and *C. albicans*), according to Rijo et al. [41]. The extracts were diluted in DMSO from 10 mg/mL to a final concentration of 1 mg/mL. Stock solutions of reference antibiotics (vancomycin, norfloxacin, and nystatin) were also prepared at 1 mg/mL in DMSO.

In aseptic conditions, Petri dishes containing 20 mL of solid Mueller–Hinton for bacteria, or *Sabouraud* Dextrose Agar culture medium (from Biokar Diagnostics), for yeasts, were inoculated with 0.1 mL of bacterial suspension matching a 0.5 McFarland standard solution and uniformly spread on the medium surface using a sterile swab. Wells of approximately 5 mm in diameter were made in the medium, using a sterile glass Pasteur pipette, and 50 μL of each extract were added into the wells. A positive control of vancomycin for Gram-positive bacteria, norfloxacin for Gram-negative bacteria, and nystatin for yeasts, and a negative control of DMSO, were used in the assay. Plates were incubated at 37 °C for 24 h. The antimicrobial activity was evaluated by measuring the diameter (mm) of the inhibition zone formed around the wells and compared to controls.

#### 3.4.3. Minimum Inhibitory Concentration (MIC) and Minimum Bactericidal Concentration (MBC)/ Minimum Fungicidal Concentration (MFC)

The MIC and MBC/MFC were determined using the microdilution technique proposed by National Committee for Clinical Laboratory Standards (NCCLS) [42]. Briefly, 100 mL of Mueller–Hilton broth for bacteria and *Sabouraud* for yeasts was placed into each well of a 96 microplate, under aseptic conditions. Each extracted sample (100 μL), the appropriate positive control of each microorganism, and negative controls at a concentration of 1 mg/mL, were added to the first well. Using a multichannel micropipette, a 1:2 microdilution series was made. A standardized bacterial suspension (10 μL), corresponding to 0.5 McFarland of each microorganism, was then placed in all wells. Finally, the plates were incubated at 37 °C for 24 h. The MIC was determined when no growth was detected in the well of the microplate. Each measurement was performed in triplicate using 96 well microtiter plates with enrichment. A total of 10 µL was withdrawn from the microplate and sown in a Petri dish to verify the MBC/MFC, which was determined when there was no visible microbial growth on the plates [43].

### 3.5. Evaluation of General Toxicity on Artemia salina Model

To evaluate the general toxicity of the different extracts and *ent*-abietane diterpene 7α-acetoxy-6β-hydroxyroyleanone, a test of lethality to *Artemia salina* (brine shrimp) was performed as described by Ntungwe et al. [31]. *A. salina* eggs, dry cyst (JBL GmbH and Co. KG, D-67141 Neuhofen Germany), were hatched in artificial seawater at 25–30 °C under aeration with a concentration of 35 g/L. A handmade container with two connected chambers was used for brine shrimp hatching. The eggs were placed in one of two compartments of a container separated by a boundary plate. The cysts were then incubated (in thermostat cabinet AQUA LYTIC^®^, Camberley, Surrey, United Kingdom) for 48 h at 24 °C. The compartment with the eggs was covered to maintain a dark ambiance. The other compartment was illuminated to attract the phototropic newly hatched nauplii through perforations on the boundary plate. The brine shrimps that had moved to the illuminated compartment were collected and used in the lethality assay. Ten to fifteen nauplii were transferred into 24-well plates containing artificial seawater, and 100 μL of each sample was added to the wells (final volume per well: 1 mL). After 24 h exposure to the samples (24 °C), the number of dead nauplii (mortality rate (%)) was determined (Equation (2)). In addition, LC_50_ (Lethal Concentration, 50%) values (μg/mL) were calculated for the most toxic extracts. DMSO was used as the solvent and was kept at 10% (*v*/*v*) in all samples tested. Potassium dichromate was used as the positive control. All samples were tested in triplicates-at a concentration of 10 ppm for each sample.
(2)$$()\mathrm{Lethal}\;\mathrm{concentration}\;\left(\%\right)=\frac{\mathrm{Total}\;A.\;\mathrm{salina}-\mathrm{Alive}\;A.\;\mathrm{salina}}{\mathrm{Total}\;A.\;\mathrm{salina}}\times 100$$

### 3.6. Cytotoxicity Screening Assays

#### 3.6.1. Cells and Cell Culture

Human colon (HCT116), breast adenocarcinoma (MCF-7), lung carcinoma (H460), and triple-negative breast cancer, MDA-MB-231S, cell lines were purchased from ATCC (Rockville, MD, USA). Cell lines were routinely cultured in 293 RPMI-1640 with ultraglutamine or DMEM (MDA-MB-231 cells) medium from Lonza (VWR, 294 Carnaxide, Portugal) supplemented with 10% fetal bovine serum from Gibco (Alfagene, Carcavelos, Portugal) and maintained in a humidified atmosphere at 37 °C with 5% CO_2_.

#### 3.6.2. Sulphorhodamine Assay

The cytotoxicity of five of the most toxic extracts on the brine shrimp lethality bioassay was carried out using the sulforhodamine B (SRB) assay as previously described [17,44,45]. Briefly, the extracts were tested in different cancer cell lines: colon colorectal carcinoma (HCT116), human breast adenocarcinoma (MCF-7), and lung cancer carcinoma (H460). Cells were plated in 96-well plates at a final density of 5.0 × 10^3^ cells/well and incubated for 24 h. Cells were then exposed to serial dilutions of each extract (from 1.56 to 50 μg/mL). The effect of the extracts was analyzed following 48 h incubation, using the sulforhodamine B (SRB) assay. Briefly, following fixation with 10% trichloroacetic acid from Scharlau (Sigma–Aldrich, Sintra, Portugal), plates were stained with 0.4% SRB from Sigma–Aldrich (Sintra, Portugal) and washed with 1% acetic acid. The bound dye was then solubilized with 10 mM Tris Base and the absorbance was measured at 510 nm in a microplate reader (Biotek Instruments Inc., Synergy, MX, USA). The solvent of the extracts (DMSO) corresponding to the maximum concentration used in these assays (0.25%) was included as a control. The concentration of extract that causes a 50% reduction in the net protein increase in cells (IC_50_) was determined for all tested extracts. Data are mean ± SEM of 4–5 independent experiments.

#### 3.6.3. MTT Assay

MDA-MB231S cell line was grown in DMEM L050-500 culture medium Biowest supplemented with 10% fetal bovine serum, L-glutamine, and penicillin-streptomycin at 37 °C and 5% CO_2_. For the MTT assay, 1000 cells/well were placed in a 96-well plate and the compound to be tested was added 24 h after sowing.

The extract and compound were prepared as stock solutions in DMSO (Scrharlau; SU01531000) at a concentration of 20 mg/mL in the case of the extract (which allowed us to use a reduced DMSO percentage at higher concentrations) and 10 mM in the case of the compound. They were stored at 4 °C. The extract and compound were prepared at different concentrations. For the extract, the following dilutions were made: 80 µg/mL, 40 µg/mL, 20 µg/mL, 10 µg/mL, and 2 µg/mL in culture medium. For the compound, the following dilutions were prepared: 10 µM, 3 µM, 1 µM, 0.3 µM, and 0.1 µM. Each concentration was assayed in triplicate in a 96-well plate.

After 48 h of treatment with the compound or extract, the cells were incubated for 2 h with MTT. After this time, the culture medium was removed, and the formazan crystals were dissolved by adding 200 µL of DMSO. The absorbance of each well was measured at 595 nm.

### 3.7. Statistical Analysis

The results were expressed as the mean value ± SD. Comparisons were performed within groups by the analysis of variance, using the ANOVA with Dunnett’s post-test. Significant differences between control and experimental groups were assessed using GraphPad Prism version 5.00 for Windows, GraphPad Software, San Diego, CA, USA, www.graphpad.com, accessed on 5 February 2021. A probability level *p* < 0.05 was considered to indicate statistical significance.

## 4. Conclusions

Natural products are known to be an important source of new anticancer agents. This study investigated the diverse biological activity of sixteen *Plectranthus* extracts. Among the studied extracts, *P. hadiensis* leaves and *P. mutabilis* had the highest percentage extraction yield, antimicrobial and antioxidant activities. *P. hadiensis* and *P. ciliatus* were the most cytotoxic extract against HCT116, MCF-7, and H460 cancer cell lines. 7α-acetoxy-6β-hydroxyroyleanone was isolated from the most cytotoxic extract (*P. hadiensis*) and was found to be 12 times more bioactive than the extract in the MDA-MB-231S cell line (triple-negative breast cancer). Therefore, it is noteworthy that 7α-acetoxy-6β-hydroxyroyleanone present in the most cytotoxic extract has interesting antitumoral activities in different cancer cell lines and might thus be responsible for the biological activity of this extract. However further phytochemical studies should be done to find out more compounds that could contribute to the cytotoxicity of *P. hadiensis*.

## Figures and Tables

**Figure 1 pharmaceuticals-14-00402-f001:**
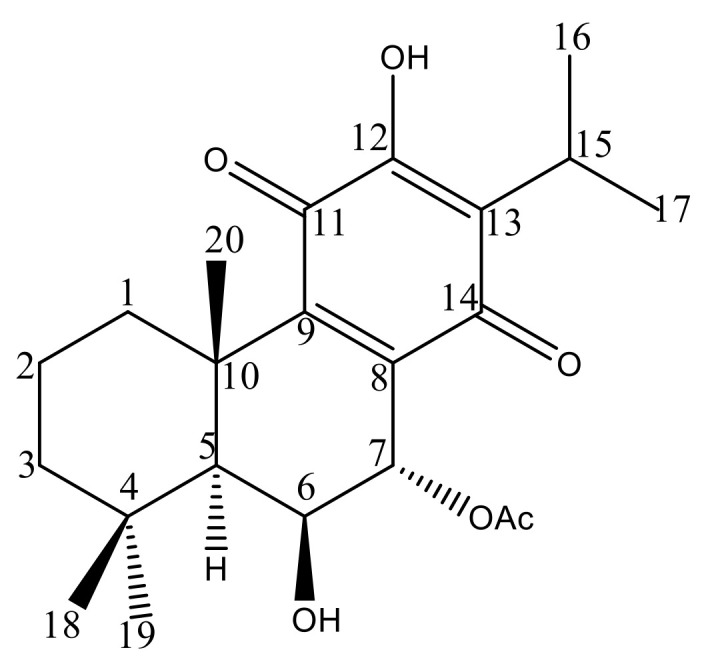
Chemical structure of 7α-acetoxy-6β-hydroxyroyleanone.

**Table 1 pharmaceuticals-14-00402-t001:** Extraction yields (dry weight % *w*/*w*), antioxidant activity, and general toxicity of sixteen *Plectranthus* spp. acetonic extracts.

Scientific Name	Yield (% *w*/*w*) ^a^	Antioxidant Activity ^b^ (%)	General Toxicity	
			* Mortality (%)	LC_50_ (µg/mL) **
*P. swynnertonii* S. Moore †	3.89	20.24 ± 0.01	65.88 ± 5	0.036 ± 1.69
*P. ciliatus* E. Mey †	11.86	13.21 ± 0.01	60.14 ± 0.44	0.504 ± 1.13
*P. mutabilis* Codd. †	30.03	46.14 ± 0.02	51.50 ± 0. 07	0.984 ± 2.92
*P. hadiensis* (Forssk.) Schweinf. Ex Sprenger †	13.49	36.24 ± 0.04	43.65 ± 3.04	0.88 ± 4.87
*P. cylindraceus* Hochst, ex Benth †	9.68	19.19 ± 0.07	43.50 ± 5.66	0.55 ± 1.96
*P. lucidus* (Benth.) Van Jaarsv. and T.J. Edwards †	6.29	23.96 ± 0.09	38.81 ± 3.75	1.053 ± 4.61
*P. inflexus* (Thunb.) Vahl ex Benth †	10.97	0.16 ± 0.05	38.70 ± 3.35	0.986 ± 2.87
*P. lippio*. Druce	2.09	30.5 ± 0.14	24.70 ± 6.22	N/A
*P. crassus* N.E.Br. †	7.77	27.27 ± 0.01	31.16 ± 1.29	N/A
*P. mzimvubuensis* Van Jaarsv. †	7.79	22.47 ± 0.05	33.95 ± 1.63	N/A
*P. xerophylus* Codd	10.16	20.15 ± 0.02	30.48 ± 3.24	N/A
*P. welshii*	2.15	15.06 ± 0.03	23.71 ± 0.60	N/A
*P. petiolaris* E. Mey ex Benth.	11.07	14.45 ± 0.01	23.42 ± 4.15	N/A
*P. woodii* Gürke	8.51	13.04 ± 0.01	29.95 ± 6.01	N/A
*P. welwitschii* (Briq. Codd)	3.59	12.63 ± 0.03	25.17 ± 5.54	N/A
*P. spicatus* E. Mey	4.75	10.57 ± 0.02	27 ± 0.28	N/A
Positive control	N/A	99.47 ± 0.10	98.89 ± 2.48	N/A
DMSO	N/A	N/A	21.87 ± 0.44	N/A

^a^ (mg of extracts/ g of the dried plant). ^b^ Antioxidant activity (%): Quercetin a potent free radical scavenging capacity was used as a positive control. Data are mean ± SD. * Screening of the *Plectranthus* spp. extracts for general toxicity at a concentration of 10 µg/mL using the *Artemia salina* test (24 h). ** LC_50_ values (µg/mL) for the most active extracts. Positive control = Potassium dichromate (10 µg/mL). Data are mean ± SD was calculated from three independent experiments and compared to DMSO († *p* < 0.001). Lethal concentration (%) = (Total *A.salina* − Alive *A.salina*)/(Total *A.salina*) × 100 was used to calculate the lethal concentration of all extracts. N/A—not applicable.

**Table 2 pharmaceuticals-14-00402-t002:** MIC and MBC/MFC (μg/mL) values of the most active acetonic extracts.

Extracts	MIC (µg/mL)			MBC/MFC (µg/mL)		
	*S. aureus*	MRSA	*C. albicans*	*S. aureus*	MRSA	*C. albicans*
Positive Control	3.91	1.95	<0.48	-	-	-
*P. mutabilis*	31.25	31.25	125	250	250	125
*P. hadiensis*	15.62	3.91	62.5	250	250	62.5

Positive controls (1 mg/mL): Gram-positive = vancomycin, Gram-negative = norfloxacin, yeast = nystatin, Negative control = DMSO, methicillin-resistant *Staphylococcus aureus* = MRSA.

**Table 3 pharmaceuticals-14-00402-t003:** IC_50_ (µg/mL) values for five selected acetonic extracts in HCT116, MCF-7, H460 and MDA-MB231S cell lines.

	HCT116 *	H460 *	MCF-7 *	MDA-MB231S **
*P. hadiensis*	3.45 ± 0.35	3.00 ± 0.10	2.90 ± 0.10	25.6
*P. ciliatus*	2.25 ± 0.75	6.45 ± 0.05	6.70 ± 0.30	N/A
*P. swynnertonii*	7.95 ± 0.35	13.50 ± 0.50	15.05 ± 0.02	N/A
*P. cylindraceus*	10.25 ± 0.75	12.50 ± 0.50	12.00 ± 1.00	N/A
*P. mutabilis*	28.00 ± 2.00	36.00 ± 2.00	35.00 ± 1.00	N/A
*Doxorubicin*	0.05 ± 3.24	0.29 ± 2.32	0.08 ± 4.10	0.07 ± 0.01

* The concentration that reduces growth by 50% (IC_50_) was determined by sulforhodamine B assay after 48 h treatment. Data are mean ± SEM of 4–5 independent experiments. ** IC_50_ (µg/mL) of the most bioactive *P. hadiensis* acetonic extract in MDA-MB231S cancer cell lines. DMSO was used as the negative control. N/A—not applicable.

## Data Availability

Not applicable.

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
