# Peer review of "Preliminary Biological Activity Screening of Plectranthus spp. Extracts for the Search of Anticancer Lead Molecules"

_pharmaceuticals, 2021, doi:10.3390/ph14050402_

Round 1

Reviewer 1 Report

The paper by Epole Ntungwe and colleagues on the biological activity of Plectranthus spp. extracts is quite interesting.

The study has a good experimental idea, is correctly designed, the methods are well described, the results are well interpreted, and the data analyzed appropriately. However, I suggest a minor revision to refine the quality of the manuscript. Authors should check the English language and correct some typos.

Author Response

The paper by Epole Ntungwe and colleagues on the biological activity of Plectranthus spp. extracts is quite interesting. The study has a good experimental idea, is correctly designed, the methods are well described, the results are well interpreted, and the data analyzed appropriately. However, I suggest a minor revision to refine the quality of the manuscript. Authors should check the English language and correct some typos.

We thank the reviewer for the comment. The quality of the paper was refined, and English language/typographical errors corrected accordingly in the manuscript.

Reviewer 2 Report

Manuscript pharmaceuticals-1193181 presents the collection and screening (antimicrobial, antioxidant, brine shrimp lethality, and cytotoxicity) of 16 species of Plectranthus that were cultivated in Lisbon, Portugal.  The manuscript is well written and the information well presented.  The results of two Plectranthus species showed promising antibacterial activity against MRSA, and several extracts showed promising cytotoxic activity against human tumor cell lines.  One of the isolated compounds showed excellent cytotoxic activity.  Publication is warranted and recommended.  I have only some minor suggestions for clarity.

Line 100:  Add a period and space after SD.

Section 4.1. Plant material:  Please indicate the location of Parque Botânico da Tapada da Ajuda.  Were the seeds, root systems, or growing plants provided by the Kirstenbosch National Botanical Gardens, South Africa?

The numbering of the sections for the Materials and Methods and Conclusions needs improvement.  A suggestion:

  1. Materials and Methods

3.1. Plant material

3.1.1. Extraction procedure

3.2. Phytochemical study of P. hadiensis

3.2.1. HPLC-DAD fingerprint analysis

3.2.2. Isolation and structural characterization of 7α-acetoxy-6β-hydroxyroyleanone

3.3. DPPH Radical Scavenging Assay

3.4. Antimicrobial Screening Assays

3.4.1. Microorganisms used

3.4.2. Well diffusion method

3.4.3. Minimum inhibitory concentration (MIC) and minimum bactericidal concentration (MBC)/ minimum fungicidal concentration (MFC)

3.5. Evaluation of General Toxicity on Artemia salina Model

3.6. Cytotoxicity Screening Assays

3.6.1. Cells and cell culture

3.6.2. Sulphorhodamine assay

3.6.3. MTT assay

3.7. Statistical Analysis

  1. Conclusions

Author Response

Manuscript pharmaceuticals-1193181 presents the collection and screening (antimicrobial, antioxidant, brine shrimp lethality, and cytotoxicity) of 16 species of Plectranthus that were cultivated in Lisbon, Portugal. The manuscript is well written, and the information well presented. The results of two Plectranthus species showed promising antibacterial activity against MRSA, and several extracts showed promising cytotoxic activity against human tumor cell lines. One of the isolated compounds showed excellent cytotoxic activity. Publication is warranted and recommended. I have only some minor suggestions for clarity

1- Line 100: Add a period and space after SD.

We thank the reviewer for the correction. We correct and changed the manuscript accordingly.

2- Plant material: Please indicate the location of Parque Botânico da Tapada da Ajuda. Were the seeds, root systems, or growing plants provided by the Kirstenbosch National Botanical Gardens, South Africa?

We thank the reviewer for the comment and we correct that the cuttings were provided by the Kirstenbosch National Botanical Gardens, South Africa. We correct and changed the manuscript accordingly.

3- The numbering of the sections for the Materials and Methods and Conclusions needs improvement. A suggestion:

  1. Materials and Methods

3.1. Plant material

3.1.1. Extraction procedure

3.2. Phytochemical study of P. hadiensis

3.2.1. HPLC-DAD fingerprint analysis

3.2.2. Isolation and structural characterization of 7α-acetoxy-6β-hydroxyroyleanone

3.3. DPPH Radical Scavenging Assay

3.4. Antimicrobial Screening Assays

3.4.1. Microorganisms used

3.4.2. Well diffusion method

3.4.3. Minimum inhibitory concentration (MIC) and minimum bactericidal concentration

(MBC)/ minimum fungicidal concentration (MFC)

3.5. Evaluation of General Toxicity on Artemia salina Model

3.6. Cytotoxicity Screening Assays2

3.6.1. Cells and cell culture

3.6.2. Sulphorhodamine assay

3.6.3. MTT assay

3.7. Statistical Analysis

  1. Conclusions

We thank the reviewer for the comment. We correct and changed the manuscript accordingly.

Reviewer 3 Report

The paper submitted by Patrícia Rijo is focused on the evaluation of natural molecules from Plectranthus spp. with potential antiproliferative, antioxidant and antibacterial activity. In particular, the active components are reported to be the abietane-type diterpenoids. The manuscript meets the standards for publication, as it is a well presented and comprehensive study. Nevertheless, some final polishing is needed before acceptance. I report some minor comments below.

  • Abstract and Introduction are both well written: concise and informative. In my opinion, introduction may me enriched by inserting a figure representing the chemical structures of the main components (possibly also from literature). Alternatively, authors could move or modify Figure 2.
  • In table 1, check the use of significant figures. In particular, see values of Mortality %: some figures/points may be missing.
  • Please check minor spelling/punctuation issues. For example, line 219: “Clifton, NJ, 218 USA.).” Check also paragraph formatting. For example, the style of titles in paragraphs 4.5 and 4.6 is not consistent. Also, the stiles of Formulas 2 and 3 do not match.
  • Conclusions: in my opinion, authors should not only focus on antiproliferative but also on antibacterial and antioxidant effect.

Author Response

The paper submitted by Patrícia Rijo is focused on the evaluation of natural molecules from Plectranthus spp. with potential antiproliferative, antioxidant and antibacterial activity. In   particular, the active components are reported to be the abietane-type diterpenoids. The manuscript meets the standards for publication, as it is a well presented and comprehensive   study. Nevertheless, some final polishing is needed before acceptance. I report some minor comments below

  1. Abstract and Introduction are both well written: concise and informative. In my opinion, introduction may me enriched by inserting a figure representing the chemical structures of the main components (possibly also from literature). Alternatively, authors could move or modify Figure 1

We thank the reviewer for the suggestion, but this is a preliminary screening of 16 Plectranthus species and hence inserting the main components of the studied spp. may be too much for the scope of this paper. Besides to the best of our knowledge, some of these spps. have little or no information in the literature. Moreover, the work to unravel the chemical composition of the second most bioactive extract (P. mutabilis) in ongoing.

  1. In table 1, check the use of significant figures. In particular, see values of Mortality %: some figures/points may be missing.

We thank the reviewer for the comment. The significant figure in table 1 was corrected in the manuscript.

  1. Please check minor spelling/punctuation issues. For example, line 219: “Clifton, NJ, 218 USA.).” Check also paragraph formatting. For example, the style of titles in paragraphs 4.5 and 4.6 is not consistent. Also, the stiles of Formulas 2 and 3 do not match.

We thank the reviewer for the comment. These were corrected in the manuscript

  1. Conclusions: in my opinion, authors should not only focus on antiproliferative but also on antibacterial and antioxidant effect.

We thank the reviewer for the suggestion. The conclusion already has the antibacterial and antioxidant activities. Here we concluded that “Among the studied extracts, P. hadiensis leaves and P. mutabilis had the highest percentage extraction yield, antimicrobial and antioxidant activities”.

This manuscript is a resubmission of an earlier submission. The following is a list of the peer review reports and author responses from that submission.

Round 1

Reviewer 1 Report

Ntungwe et al describe the preliminary characterization of acetonic extracts from 16 Plectranthus species. They analyze the antimicrobial, antifungal, cytotoxicity and anticancer activities and identify two extracts with more activity. Moreover, they characterize the most bioactive extract further and identify one component that is most likely the active ingredient. Although it may not be sufficiently active to serve itself as a drug, it could be used as a lead compound that can be modified to be more active.

Major comments

Table 1. The authors use quercetin as a positive control and express the activity of the acetonic extract in percentage of the quercetin activity. This is somewhat misleading, since quercetin is pure, whereas the extracts not, it appears that apples are compared to oranges… A statement to address this should be added.

Table 1. When preparing these extracts, how is the repeatability? Does every time and extract was made the same result is obtained? Some statement should be added that the extraction has been made several times with similar yield and activity, and the discrepancy between the different species is not due to extraction variability.

Line 131 and 188: Artemia salina model is the same as Brine shrimp lethal bioassay?

Line 155: …and thus is difficult to treat

Line 188 and 369 and Figure legend S4: Roy...the full name should be used

Line 212: the Plant List… which list? Reference?

Line 217: This procedure was repeated … not clear what was used to repeat, the filtrate?

Line 286: 100 microL

Reviewer 2 Report

The authors tested the bioactivity of a series of extracts from Plectranthus spp. and concluded that the most active one mainly contains 7α-acetoxy-6β-hydroxyroyleanone which has already been showed as an effective molecule in different cancer cell lines.

The manuscript is quite difficult to read in some passages, as it seems to lack a flux of information (see below for details).

Major concerns:

Results and discussion. A brief description of the extraction method would have been helpful, instead of only redirecting to a reference. The same is valid also for assays used along the paper.

Table 1. The mention in the text to the General toxicity data is not “so close” to the table, so it would be better to state something about that in the legend of the table. on how they were calculated. There’s no mention in the text of this part of the table.

How could the negative control (DMSO) be considered as negative with 20% of mortality? And so, which is the statistical significance of the data referring to the different extracts?

Page 3, Lines 115-116. The authors stated “As the initial screening identified P. hadiensis and P. mutabilis extracts as possessing the most promising antibacterial activities” on which bases this is decided? Looking at table 1 P. swynnertonii and P. ciliatus seems to be more effective.

Table 2. What is MRSA?

Page 4, line 130. It is not clear which is the experimental workflow, as here it is stated that the artemia salina test has been performed to identify the toxicity, which is reported in table 1, which give rise to the fact that the most promising extract were analysed for antibacterial activity, described in page 3.

Table 3. Why expressing. Those values as GI50 and IC50? which would be the difference? And why for IC there is µl/ml and for GI not? Moreover, why the authors do not perform an MTT assay for all the cell lines tested, as to obtain more uniform data? Which are the negative controls for the two assays?

Page 4, line 143: What is the meaning of “…crude extracts that possess GI50 in the range ≤ μg/mL…”? It is a range or a value less than?

Page 5, lines 181-184. Please name the cell lines.

Page 5, line 188. Please define Roy at its first appearance and not only in the conclusion.

Page 8, line 325, 326. What’s the meaning of serum 295 and 296 incubator?

Page 9, MTT Assay. Why the extract was suspended at 2 mg/ml instead of 10 mg/ml, as stated at the beginning of the methods? If this is the case the reported IC50 of 25,6 µg/ml corresponds to a dilution of approx. 1:80, which is the value of the same DMSO dilution?